

# Search for ultralight axion dark matter in a side-band analysis of a ¹⁹⁹Hg free-spin precession signal

C. Abel[1], N. J. Ayres[1,2], G. Ban[3], G. Bison[4], K. Bodek[5], V. Bondar[2,4,6],
E. Chanel[7], C. B. Crawford[8], M. Daum[4], B. Dechenaux[3], S. Emmenegger[4],
P. Flaux[3], W. C. Griffith[1], P. G. Harris[1], Y. Kermaidic[9†], K. Kirch[2,4], S. Komposch[2,4],
P. A. Koss[6‡], J. Krempel[2], B. Lauss[4], T. Lefort[3], P. Mohanmurthy[2,4],
O. Naviliat-Cuncic[3], D. Pais[2,4], F. M. Piegsa[7], G. Pignol[9], M. Rawlik[2∘], D. Ries[10],
S. Roccia[9], D. Rozpedzik[5], P. Schmidt-Wellenburg[4⋆], N. Severijns[6], Y. V. Stadnik[11],
J. A. Thorne[1,7], A. Weis[12], E. Wursten[6§], J. Zejma[5] and G. Zsigmond[4]

**1** Department of Physics and Astronomy, University of Sussex, Falmer, Brighton BN1 9QH, UK
**2** ETH Zürich, Institute for Particle Physics and Astrophysics, CH-8093 Zürich, Switzerland
**3** LPC Caen, ENSICAEN, Université de Caen, CNRS/IN2P3, Caen, France
**4** Paul Scherrer Institut, CH-5232 Villigen PSI, Switzerland
**5** Marian Smoluchowski Institute of Physics, Jagiellonian University, 30-059 Cracow, Poland
**6** Instituut voor Kern- en Stralingsfysica, University of Leuven, B-3001 Leuven, Belgium
**7** Laboratory for High Energy Physics and Albert Einstein Center for Fundamental Physics, University of Bern, CH-3012 Bern, Switzerland
**8** University of Kentucky, 40506 Lexington, USA
**9** Univ. Grenoble Alpes, CNRS, Grenoble INP, LPSC-IN2P3, Grenoble, France
**10** Department of Chemistry - TRIGA site, Johannes Gutenberg University Mainz, 55128 Mainz, Germany
**11** School of Physics, The University of Sydney, NSW 2006, Australia
**12** Physics Department, University of Fribourg, CH-1700 Fribourg, Switzerland

⋆ philipp.schmidt-wellenburg@psi.ch

## Abstract

**Ultra-low-mass axions are a viable dark matter candidate and may form a coherently oscillating classical field. Nuclear spins in experiments on Earth might couple to this oscillating axion dark-matter field, when propagating on Earth's trajectory through our Galaxy. This spin coupling resembles an oscillating pseudo-magnetic field which modulates the spin precession of nuclear spins. Here we report on the null result of a demonstration experiment searching for a frequency modulation of the free spin-precession signal of ¹⁹⁹Hg in a $1\,\mu$T magnetic field. Our search covers the axion mass range $10^{-16}$ eV $\lesssim m_a \lesssim 10^{-13}$ eV and achieves a peak sensitivity to the axion-nucleon coupling of $g_{aNN} \approx 3.5 \times 10^{-6}$ GeV$^{-1}$.**

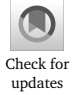

---

† Present address: Université Paris-Saclay, CNRS/IN2P3, IJCLab, 91405 Orsay, France
‡ Present address: Fraunhofer-Institut für Physikalische Messtechnik IPM, 79110 Freiburg i. Breisgau, Germany
∘ Present address: aul Scherrer Institut, CH-5232 Villigen PSI, Switzerland
§ Present address: RIKEN, Ulmer Fundamental Symmetries Laboratory, 2-1 Hirosawa, Wako, Saitama, 351-0198, Japan

# 1 Introduction

The existence of cold dark matter (DM) is a cornerstone of the cosmological standard model [1], while the standard model of particle physics (SM) [2] convincingly describes all laboratory-known fundamental constituents of matter and their interactions without providing a candidate particle for DM. The discovery of the microscopic constituents and fundamental interactions of DM, which makes up about 85% of the matter content of the Universe, and how it fits into the SM would resolve one of the most intriguing riddles of modern science.

One very well-motivated category of candidates for cold DM is the class of axion-like particles (ALP). The prototype of these pseudoscalar bosons, the canonical axion, was first introduced by Peccei and Quinn [3] to resolve the strong CP problem in quantum chromodynamics (QCD) [4–10]. More generic ALPs are spin-0 bosons where the stringent relation between coupling and mass required to solve the QCD CP problem is relaxed. Such ALPs and other ultra-low-mass bosons can be incorporated into many beyond-SM theories and can provide a sufficient abundance to match the observed DM abundance [11–17]. In the remainder of this article, we will follow the general taxonomy and simply refer to these DM candidates as axions.

The wide range of testable parameters and the numerous theoretical motivations of axions coupling to SM particles instigated a large variety of experimental searches and proposals in the last decade [18–28]. These searches exploit three possible types of non-gravitational interactions which lead to distinctive phenomena: (1) the coupling to photons, e.g., in searches for axions via conversion to photons in microwave cavities [19], searches for an oscillating-in-time magnetic field [29,30], detection of axions emitted by the Sun [31], or light-shining-through-a-wall experiments [32,33]; (2) the coupling to gluons inducing an oscillating electric dipole moment [21,34,35]; and (3) the coupling to fermion spins, also known as the axion-wind effect [36,37], resulting in pseudo-magnetic spin precession [21,23–25,28].

In this article, we present the result from a search for the axion-wind effect exploiting the spin precession of polarized $^{199}$Hg in a low magnetic field of $B = 1\,\mu$T, using the same apparatus as in the ultra-low-mass axion DM search, $m_a \lesssim 10^{-17}$ eV, for the axion-gluon and axion-nucleon couplings [21]. The presented search covers the axion mass range $10^{-16}$ eV $\lesssim m_a \lesssim 10^{-13}$ eV, exceeding the sensitivity of the limit from a comagnetometer measurement using $^{13}$C-formic acid [24], but less sensitive than the result using historical data from a $^{3}$He/K-comagnetometer [38,39] in the same mass range, if one assumes that the axion field couples to proton and neutron spins equally. However, our $^{199}$Hg system is predominantly sensitive to the axion-neutron interaction, whereas the systems in Refs. [24,38] are mainly sensitive to the axion-proton interaction, providing complementarity if the axion field couples to protons and neutrons with different strengths. Furthermore, the local DM density may vary significantly from one axion coherence time to another or due to possible clumping of DM on sub-galactic length scales, so measurements taken at different times can nevertheless provide useful complementary information even when the axion-proton and axion-neutron coupling strengths are equal.

Our axion-wind search and result is based on the assumption that Earth passes through DM with an orbital velocity of $v_0 \approx 10^{-3}c$, where $c$ is the speed of light in vacuum. DM particles with a mass below $10\,\text{eV}/c^2$ must be bosons if DM dominantly consists of only one type of particle. This is a direct consequence of the average local DM density being inferred to lie in the $2\sigma$ range $\rho_{\text{DM}} = 0.4 - 0.8\,\text{GeV/cm}^3$ [40] and the requirement of a large number of particles residing within a single de Broglie volume. In this hypothesis, the DM made of axions of mass

$m_a$ can be described as a classical field,

$$a(t) = a_0 \sin(\omega_a t) \,, \tag{1}$$

where $a_0$ is the oscillation amplitude at the Compton frequency $\omega_a = m_a c^2/\hbar$, and $\hbar$ is the reduced Planck constant. In this article, we set $\hbar = c = 1$ unless stated otherwise. The field amplitude,

$$a_0 = \frac{\sqrt{2\rho_{\mathrm{DM}}}}{\omega_a} \,, \tag{2}$$

is related to the average local DM density for which we take the lower conservative bound of $\rho_{\mathrm{DM}} = 0.4\,\mathrm{GeV/cm^3}$ for the $2\sigma$ range quoted above Eq. (1). The coherence time of the axion DM field is governed by the characteristic spread in axion kinetic energies and is given by $\tau_c = 2\pi/\Delta E_a \sim 10^6 T_{\mathrm{osc}}$, where $T_{\mathrm{osc}} \approx 2\pi/m_a$ is the DM period of oscillation.

The interaction of the axion with fermions is described by the Lagrangian (in the notation of, e.g., [22]):

$$\mathcal{L}_a = g_{a\psi\psi} \partial_\mu a \bar{\psi} \gamma^\mu \gamma^5 \psi \,, \tag{3}$$

where $\psi$ and $\bar{\psi} = \psi^\dagger \gamma^0$ are the fermion field and Dirac adjoint, respectively, e.g., of the nucleons inside the $^{199}$Hg nucleus, and $g_{a\psi\psi}$ is the coupling constant between the axion field and the fermion. In our case for the spin-1/2 $^{199}$Hg nucleus, this interaction can be expressed by the following non-relativistic Hamiltonian if the axion field couples to neutron and proton spins with equal strength (see, e.g., [37] for details),

$$H_a = \hbar g_{\mathrm{aNN}} \vec{\sigma} \cdot \vec{D} \,, \tag{4}$$

describing the coupling of the $^{199}$Hg nuclear spin to the axion DM field. In analogy to Larmor precession, this leads to a precession of the $^{199}$Hg nuclear spin in the pseudomagnetic field $\vec{D} \approx a_0 \vec{p} \sin(m_a t)$, where the gyromagnetic ratio $\gamma_{\mathrm{Hg}}$ is substituted by the coupling parameter $g_{\mathrm{aNN}}$, and $\vec{p} \approx m_a \vec{v}$ is the axion DM momentum relative to the detector.

In our analysis, we assume that the instantaneous velocity $\vec{v}$ is directed opposite to the orbital velocity of Earth with respect to the Galactic Center, $\vec{v}_0$, which is expected to be closely aligned with the time-average velocity of Earth relative to the DM halo locally, $\langle \vec{v} \rangle$. The value of $\vec{v}$ at some point in time and space generally differs from $\langle \vec{v} \rangle$ due to the DM's non-zero virial velocity $\vec{v}_{\mathrm{vir}}$. In the event of non-detection of a DM signal, accounting for such possible stochastic fluctuations in the direction and amplitude of $\vec{v}$ leads to practically the same limits as assuming the fixed value $\vec{v} = \langle \vec{v} \rangle$ when the magnitudes of $\vec{v}_0$ and $\vec{v}_{\mathrm{vir}}$ are comparable [41] (for earlier attempts to account for such stochastic fluctuations, see [42]). Hence in our analysis, we simply assume $\vec{v} = \vec{v}_0$, as was done in Refs. [21, 23–25].

The $^{199}$Hg magnetometer measures the magnetic-field strength $B$ by observing the precession frequency $\omega_{\mathrm{Hg}} = \gamma_{\mathrm{Hg}} B$, which is the Larmor frequency of the nuclear spin, using the spin-orientation-dependent absorption of circularly polarized light resonant with the $^{199}$Hg $6\,^1S_0 \rightarrow 6\,^3P_1\ F = 1/2$ transition. The detected light power $\mathcal{P}_1(t)$ after transmission through polarized $^{199}$Hg vapor as a function of time, $t$, can be described as

$$\mathcal{P}_1(t)/\mathcal{P} = e^{-\mathcal{O}_{\mathrm{Hg}}(t)[1 - P(t)\sin(\omega_{\mathrm{Hg}} t + \phi)]} \,, \tag{5}$$

where $\mathcal{P}$ is the initial light power, $\mathcal{O}_{\mathrm{Hg}}(t) = n(t)\sigma_{\mathrm{Hg}} \ell$ is the opacity of the vapor depending on the vapor density $n(t)$, the light absorption cross section $\sigma_{\mathrm{Hg}}$, and length $\ell$ of the light path. The vapor polarization, $P(t) = P_0 \exp(-\Gamma_2 t)$, is a function of the initial polarization $P_0$ and transverse spin relaxation time $T_2 = 1/\Gamma_2$. The vapor density $n(t) = n_0 \exp(-\Gamma_\ell t)$ decreases with time due to the chamber-specific leakage rate $\Gamma_l$, a result of mechanical imperfections. For now we ignore the effect of depolarization and vapor leakage, which in practice result in a

broadening of the resonant frequency. Hence, in the presence of an oscillating axion field we will observe a frequency-modulated light power of the form

$$\mathcal{P}_{\mathrm{FM}}(t) = \mathcal{P}_0 \cos\left[\gamma_{\mathrm{Hg}} B t + \frac{1}{2} \int_0^t g_{\mathrm{aNN}} \nu \sqrt{2\rho_{\mathrm{DM}}} \sin(\omega_a t') \hat{\vec{B}} \cdot \hat{\vec{p}}(t') \, \mathrm{d}t'\right], \tag{6}$$

where $\mathcal{P}_0$ is the oscillation amplitude. The function $\hat{\vec{B}} \cdot \hat{\vec{p}}(t)$ arises due to Earth's rotation (which changes the angle between the directions of the applied magnetic field and the axion DM flux) and is given by

$$\hat{\vec{B}} \cdot \hat{\vec{p}}(t) = \cos(\chi)\sin(\delta) + \sin(\chi)\cos(\delta)\cos(\Omega_{\mathrm{sid}} t - \eta), \tag{7}$$

where $\chi$ is the angle between Earth's axis of rotation and the spin quantization axis [$\chi = 42.5°$ at the location of the Paul Scherrer Institute (PSI)], $\delta \approx -48°$ and $\eta \approx 138°$ are the declination and right ascension of the Galactic axion DM flux relative to the Solar System [21], respectively, and $\Omega_{\mathrm{sid}} \approx 7.29 \times 10^{-5}$ s$^{-1}$ is the daily sidereal angular frequency. For a weak DM-induced modulation, this can be reduced to a superposition of three amplitude oscillations

$$\mathcal{P}_{\mathrm{FM}}(t) \approx \mathcal{P}_0 \Big[ \cos\left(\gamma_{\mathrm{Hg}} B t\right) \tag{8}$$
$$+ \hat{\vec{B}} \cdot \hat{\vec{p}}(t) \frac{g_{\mathrm{aNN}} \nu \sqrt{2\rho_{\mathrm{DM}}}}{2\omega_a} \cdot \left[ \sin\{(\gamma_{\mathrm{Hg}} B + \omega_a) t + \phi_+\} + \sin\{(\gamma_{\mathrm{Hg}} B - \omega_a) t + \phi_-\}\right]\Big],$$

using a series expansion to first order in Bessel functions of the first kind [43], and noting that $\Omega_{\mathrm{sid}} \ll \omega_a$ for the axion masses considered in our present search. In the frequency domain, this results in a peak with amplitude $\mathcal{P}_0$ and two sideband peaks of amplitude $\mathcal{P}_0 g_{\mathrm{aNN}} \nu \sqrt{2\rho_{\mathrm{DM}}}/(2\omega_a)$. For axion masses with oscillation frequencies $m_a c^2/\hbar \approx \omega_a < \omega_{\mathrm{Hg}}$, we expect two narrow lines at $\omega_{2p} = \omega_{\mathrm{Hg}} \pm \omega_a$, while for $\omega_a > \omega_{\mathrm{Hg}}$ we search for two lines at $\omega_{2p} = \omega_a \pm \omega_{\mathrm{Hg}}$. Note that since $\Omega_{\mathrm{sid}} \ll 1/T_2$ (see Fig. 2), the additional sidereal sideband peaks induced by Earth's rotation are not resolvable in this experiment and we can treat $\hat{\vec{B}} \cdot \hat{\vec{p}}(t)$ as a slowly time-varying function.

## 2 Experimental setup

The measurements were performed in 2017 using the same instrument which was used for data taking for the most sensitive measurement of the static electric dipole moment (EDM) of the neutron [44]. Figure 1 shows a sketch of the apparatus. The mercury isotope $^{199}$Hg was used to measure the Larmor precession frequency $f_{\mathrm{Hg}} = \gamma_{\mathrm{Hg}}\langle B \rangle$, where $\gamma_{\mathrm{Hg}} = 7.590\,115\,2(62)\,\mathrm{MHz/T}$ [45] is the gyromagnetic ratio of $^{199}$Hg and $\langle B \rangle$ is the volume-averaged magnetic field within the precession chamber. The cylindrical precession chamber of height $h = 120\,\mathrm{mm}$ and radius $r = 235\,\mathrm{mm}$ was made of diamond-like-carbon-coated [46] aluminum top and bottom plates and an insulator ring made from polystyrene coated with deuterated polystyrene [47]. Two circular windows of about 50 mm diameter inside the insulator ring were made of quartz glass coated with deuterated polyethylene [47] and were used to shine a laser beam through the chamber to detect the precession signal. A detailed description of the mercury magnetometer and its laser upgrade can be found in Refs. [48–50]. Polarized mercury was prepared inside a cylindrical volume of 1 L just below the bottom plate. A shutter with a 12 mm-wide diameter separated the polarizing chamber for optical pumping from the larger volume of the precession chamber. A single measurement, which we call a cycle, consisted of the following steps: firstly, the shutter between the polarizing and precession chambers opened for $t = 2\,\mathrm{s}$ to admit polarized $^{199}$Hg vapor into the detection volume. Next,

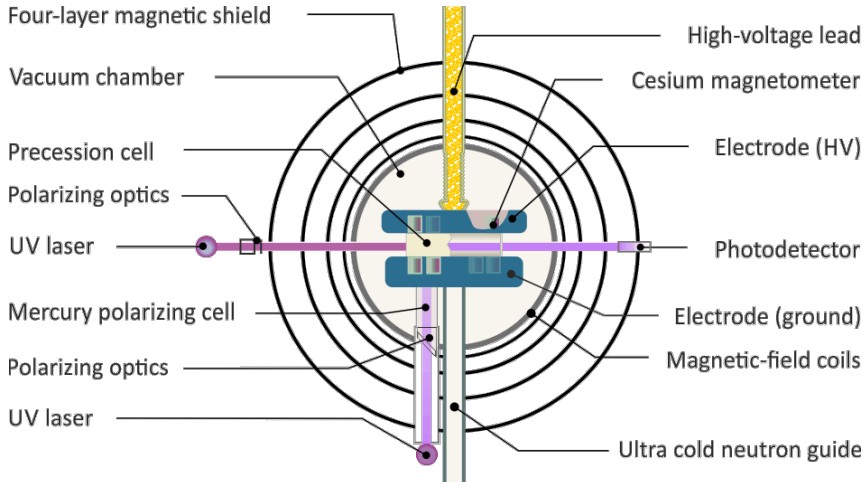

Figure 1: Sketch of the single chamber nEDM spectrometer used for the measurement, omitting details relevant only for measurements with ultracold neutrons.

an optimized circularly rotating magnetic field $B_{rf}$ was applied during $t_{rf} = 2\,$s using two split coils perpendicular to each other and to the magnetic field $\vec{B}$ with frequency $f_{rf} = f_{Hg} \approx 7.88\,$Hz. The coils were wound onto the vacuum tank made of aluminum. The currents $I_i$ generating $B_{rf}(I_i)$ were adjusted such that $\gamma_{Hg}B_{rf}(I_i)t_{rf} = \pi/2$, tilting the spin into the plane perpendicular to the magnetic field $\vec{B}$. While one batch of polarized mercury was prepared by optical pumping [50], the other was precessing inside the precession chamber. The initial $^{199}$Hg-vapor pressure was about $4.4 \times 10^{-7}$ mbar, calculated from the initial opacity, shown in Fig. 2, with the $^{199}$Hg absorption cross section of $2 \times 10^{-13}\,$cm$^2$. The spin-dependent light absorption of the $^{199}$Hg nuclei results in an power modulation at the Larmor frequency, $f_{Hg}$, of the circularly-polarized readout laser light.

The light power of the readout beam was recorded using a photo multiplier (PM). The signal from the PM was digitized using a 24-bit $\Delta\Sigma$ analog-to-digital converter (ADC) at a sampling frequency of 51.2 kHz of the National Instrument PXI-4461 module. The data were down-sampled to $f_s = 100\,$Hz using a spline interpolation before it was saved to disc with an absolute time stamp of 1 ms precision. The precise timing of spin-flip pulses and ADC-sampling was controlled by an atomic clock with a relative precision of $1 \times 10^{-12}$.

## 3 Data analysis

A total of 297 free spin-precession (FSP) signals were recorded in a little more than $T \approx 13$ hours. Each FSP is a time series of 10,000 data points recorded with a frequency of 100 Hz and a duration of $\mathcal{T} = 100\,$s. In order to estimate the transmitted light power $\mathcal{P}$, opacity $\mathcal{O}$, and polarization $P = (P_1^2 + P_2^2)^{1/2}$, we modified equation (5) to

$$
\begin{aligned}
\mathcal{P}_1(t) = \mathcal{P}\exp\big[ &-\mathcal{O}\exp\left(-\Gamma_\ell t\right) \\
&+ \mathcal{O}\exp\left\{-\left(\Gamma_\ell + \Gamma_2\right)t\right\}\left\{P_1\sin\left(\omega_{Hg}t\right) + P_2\cos\left(\omega_{Hg}t\right)\right\} \\
&+ \mathcal{O}\exp\left\{-\left(\Gamma_\ell + \Gamma_2 + \Gamma_{NL}\right)t\right\}\left\{P_3\sin\left(2\omega_{Hg}t\right) + P_4\cos\left(2\omega_{Hg}t\right)\right\}\big] ,
\end{aligned} \tag{9}
$$

to which we fitted the FSP time series. In the second step, we also include the second-harmonic terms in the final line of Eq. (9), which appear due to non-linear effects in the detection scheme. While we kept $\mathcal{P}$ fixed to the values obtained in the first optimization, we introduced the additional parameters $\Gamma_{NL}$ and $P_{NL} = (P_3^2 + P_4^2)^{1/2}$ for the second-harmonic terms. The results

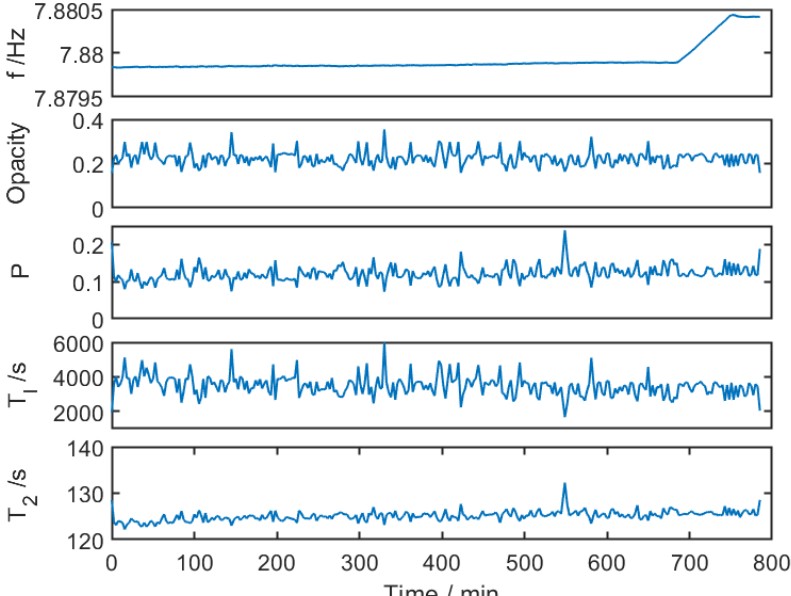

Figure 2: Results from the fit of Eq. (9) to all FSP signals versus time in minutes. Plots show from top to bottom: the mercury Larmor frequency $f_{\text{Hg}}$, the initial opacity $\mathcal{O}$, the spin polarization $P$, the leakage time constant $T_\ell$, and the transverse decoherence time constant $T_2$. Note that the extracted Larmor frequencies, in the upper most panel, are all well within 1 mHz, although the ramp of a strong dipole magnet in the near vicinity of the experiment is visible at times exceeding 660 min.

of the principal parameters are shown for the entire dataset in Figure 2. In the case of low opacity, the remaining residual times series can be expanded to first order in $\mathcal{O}$,

$$R(t) = \mathcal{N}(t) + \hat{\vec{B}} \cdot \hat{\vec{p}}(t) \; \frac{g_{\text{aNN}} v \sqrt{2\rho_{\text{DM}}}}{2\omega_a}$$
$$\mathcal{P}_0 \mathcal{O} e^{-(\Gamma_\ell + \Gamma_2)t} \left[ \sin\left\{ \left(\gamma_{\text{Hg}}B + \omega_a\right)t + \phi_+ \right\} + \sin\left\{ \left(\gamma_{\text{Hg}}B - \omega_a\right)t + \phi_- \right\} \right], \quad (10)$$

and contains the noise $\mathcal{N}(t)$ and the hypothetical frequency modulation at $\omega_a$, while neglecting the sidereal modulation due to the limited resolution. Averaging the relative orientation of the magnetic field with respect to the Milky Way over the data-taking period in 2017, starting on June 28 at 17h31m and ending on June 29 at 6h39m, results in $\langle \hat{\vec{B}} \cdot \hat{\vec{p}} \rangle \approx 0.7$ for $\vec{p}$ directed opposite to the orbital velocity of Earth with respect to the Galactic Center. An axion DM field would result in coherent modulations of the FSP signals across all time-series for frequencies $\omega_a/(2\pi) = f_a \lesssim 10^{-6}/T \approx 21.4$ Hz.

The residuals of all 297 cycles were stitched together into one long time-series using the timestamps of the atomic clock. We analyzed the macro time-series using Lomb-Scargle's periodogram method [51, 52], which is ideally suited for a frequency analysis of non-equally spaced data. This results in a power spectrum with $i = 1 \ldots 9443686$ entries $\mathcal{P}_{\text{LS},i} = A_i^2 + B_i^2$ giving the power of a periodic signal at a frequency $f_i$ with sine and cosine amplitudes of $A_i$ and $B_i$, respectively, spaced by $\delta f = 1/(4T) = 5.3 \times 10^{-6}$ Hz.

A no-signal background hypothesis was constructed by assuming only a single axion field and using the noise floor to estimate the background. Hence, the measured noise floor in the direct vicinity of a peak represents the expected background in this frequency range. The noise floor, $C_i = \left( \sum_{i-1000}^{i+1} \mathcal{P}_{\text{LS}}(f_i) + \sum_{i+1}^{i+1000} \mathcal{P}_{\text{LS}}(f_i) \right)/2000$, was estimated by averaging over a frequency range $\pm 5.3$ mHz, and removing outliers, peaks above the $3\sigma$ level. The sensibility of

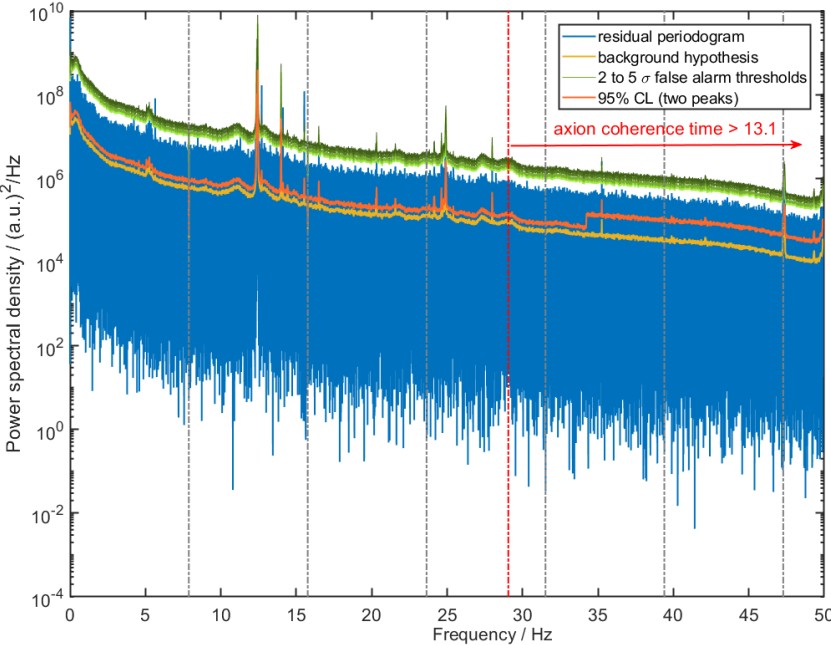

Figure 3: Lomb-Scargle power spectral density $\mathcal{P}_{\mathrm{LS}}(f_i)$ [51,52] of fit residuals. The 95% confidence level is calculated for a two-peak signal assumption. False-alarm thresholds, from $2\sigma$ to $5\sigma$, are indicated for a single axion frequency assumption. Several spurious peaks pass the false alarm threshold, but none of them have a corresponding second modulation peak and hence cannot originate from a magnetic or pseudo-magnetic coupling to the $^{199}$Hg-spin. The dashed gray vertical lines at multiples of 7.88 Hz indicate the positions of the mercury Larmor frequency and higher orders. The red dash-dotted line indicates the frequency above which the expected coherence time of the axion field would be shorter than our measurement time.

the data analysis was cross-checked by numerically generating a similar data set with signal, based on the experimental background noise spectrum by adding a frequency modulation to the $^{199}$Hg precession.

To calculate the local $p$-value for the power $\mathcal{P}_{\mathrm{LS}}(f_i)$ of the $i$th frequency, we take advantage of the fact that the cumulative distribution function of a white noise power spectrum follows an exponential distribution, hence

$$p_{\mathrm{local}}(f_i) = \exp\left[-\mathcal{P}_{\mathrm{LS}}(f_i)/\langle B_i\rangle\right] . \tag{11}$$

Figure 3 shows the Lomb-Scargle frequency analysis of the residuals with the 95% confidence level from the single-axion hypothesis assuming the presence of two peaks. The lowest accessible side band frequency is $f_{\mathrm{min}} = \pi/T_2$ defined by the decoherence time $T_2$ of the carrier signal. At frequencies $f_s/2 - 2f_{\mathrm{Hg}} \geq 34.24$ Hz, cases in which $f_s/2 - f_{\mathrm{Hg}} \leq f_a$, only one of two peaks would be detectable. Hence, the 95% C.L. threshold increases. However, for this axion search we only consider data taken up to the inverse of the expected axion coherence time $\tau_c$. We assume that the local $p$-values at different trial frequencies are uncorrelated and therefore calculate the global $p$-value according to [53]:

$$p_{\mathrm{global}} = 1 - \left(1 - p_{\mathrm{local}}\right)^n , \tag{12}$$

where $n$ is the number of trial frequencies. This results in the indicated false-alarm thresholds for $2\dots5\sigma$ significance as shown by the green lines in Fig. 3.

# 4 Result and Discussion

No unambiguous sideband signal arising from a frequency modulation due to an oscillating ultralight axion dark matter field was found in the periodogram shown in Fig. 3. Although several spurious peaks pass the false alarm thresholds, none of them have a corresponding second modulation peak, and hence cannot origin from a magnetic or pseudo-magnetic coupling to the $^{199}$Hg-spin. We assume that these single peaks are either resonances in the laser stabilization scheme which was used, e.g. the peak at 12.5 Hz was also present in commissioning data [50] taken without mercury vapor, or a result of the analog and digital DAQ system in use. As the experiment was dismounted in late 2017 it was not possible to track back the origin of these spurious outliers. Hence, no signal consistent with axion DM was observed and we translate the Lomb-Scargle periodogram into a 95% confidence limit on the coupling constant $g_{aNN}$, between the axion field and the nucleons within the $^{199}$Hg nucleus by calculating

$$g_{aNN,95\%}(f_a) = A_{osc,95\%}(f_i) \frac{4\pi \left(f_i - f_{Hg}\right)}{v \sqrt{2\rho_{DM}}\langle I_0 \rangle \exp\left(-100\,\text{s}/\langle \tau \rangle\right)}, \tag{13}$$

where $f_a = f_i - f_{Hg}$ is the trial axion frequency, $\langle \tau \rangle = \langle \Gamma_l + \Gamma_2 \rangle^{-1} = 121.5(5)\,\text{s}$ is the mean decay time constant of the mercury precession signal with mean frequency $f_{Hg} = 7.879(1)\,\text{Hz}$ and mean initial oscillation amplitude $\langle I_0 \rangle$, averaged over all cycles. The oscillation amplitude, $A_{osc,95\%}$, is calculated for a coherent signal from the 95% confidence band of the power spectral density with $A_{osc,95\%} = \sqrt{P_{LS,95\%}}$.

The result of our search for the coupling of axion DM to $^{199}$Hg nucleon spins is shown in Figure 4. The insensitive regions at multiples of the carrier frequency $f_{Hg} = 7.879(1)\,\text{Hz}$ are not depicted. The dip in sensitivity around 4 Hz results from the broad peak-like noise structure around 12.5 Hz in the Lomb-Scargle spectrum. The red shaded area, labeled "PSI-HgM-sideband", indicates the upper bound on the axion coupling to nucleon spins at the 95% confidence level in the examined frequency range. The sensitivity represents 297 free spin decay signals of 100 s duration in a magnetic field of $B = 1038\,\text{nT}$ taken within 13 hours of operation. Systematic effects due to the readout scheme, e.g., the real-light-shift systematic discussed in Ref. [44], are tiny with respect to the resolution and only affect the absolute value of the carrier frequency.

Figure 4 assumes the case when the axion-neutron and axion-proton interaction parameters are equal, $g_{aNN} = g_{aPP}$. Our $^{199}$Hg system is primarily sensitive to $g_{aNN}$ with a factor of $\approx 10$ weaker dependence on $g_{aPP}$ [54], whereas the systems in Refs. [24, 38] are mainly sensitive to $g_{aPP}$. The limit derived from historical data from a $^3$He/K-comagnetometer at Princeton has been analyzed first by the team of group Bloch et al. [38] using a published power spectrum [55] and later re-analyzed by the Princeton group [39] using the original raw data. Both approaches (we show the limit from [39]) give a similar limit shown by the light turquoise region. The $^{129}$Xe system in Ref. [56] is mainly sensitive to $g_{aNN}$, but with a factor of $\approx 3$ weaker dependence on $g_{aPP}$ [54]. Hence when $g_{aNN} \neq g_{aPP}$, our experiment provides a probe of axion-nucleon interactions that is complementary to the other searches.

Using the new double-chamber neutron EDM apparatus [57] currently being setup at PSI will permit an increase in intrinsic sensitivity by a factor of at least 10. By prolonging the free precession time to 240 s, twice the coherence time, we will increase the effective data taking time by a factor of 2.4, while a differential detection scheme will be used to cancel and hence reduce the intrinsic noise by at least a factor of 2. The statistical sensitivity of the $^{199}$Hg atoms will be increased by a factor of 2 using two precession chambers set one above the other, each with a two times larger volume. In addition, we plan to take data for at least 10 days, nearly 20 times the shortest axion coherence times of this measurement. This will sample the stochastic behavior of the coherent DM field and result in an improved sensitivity by a factor of $20^{1/4} \approx 2$.

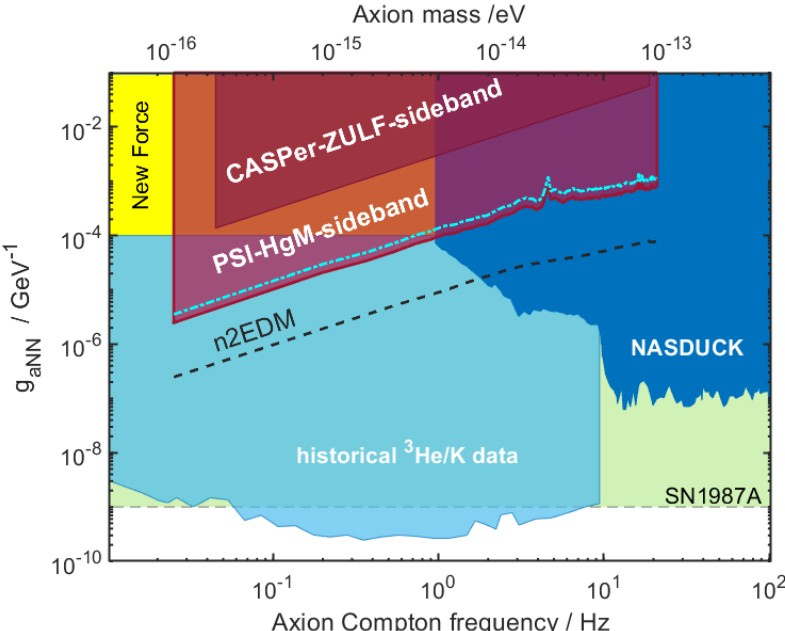

Figure 4: Exclusion plot for the axion-nucleon coupling parameter $g_{aNN}$ versus Compton frequency of the classical axion field $f_a \approx m_a/h$, assuming that the axion field couples to neutrons and protons with the same strength. The red area excluded by our present $^{199}$Hg-sideband analysis at PSI with a 95% C.L. supersedes entirely the previous work by CASPEr-ZULF [24] (brown region, 90% C.L.). The dashed-dotted cyan line indicates the limit when taking the average orientation of the quantization axis relative to the Galaxy, $\langle \hat{\vec{B}} \cdot \hat{\vec{p}} \rangle \approx 0.7$, into account. The dashed black line indicates the prospects of a similar measurement using the new apparatus for the search of a neutron EDM currently under construction at PSI [57]. The turquoise region, marked as "historical $^3$He/K data", is the 95% C.L. limit derived from helium-potassium co-magnetometer data [38,39] (we show the limit from [39] derived using the original raw data), while the blue region represents the 95% C.L. limit from the NASDUCK experiment [56]. The yellow "new force" region is a limit from a search for long-range spin-dependent force [55], while the green region marked "SN1987A" represents astrophysical limits from the cooling of supernova 1987A [58], with both of these limits being independent of the axion contribution to the local dark-matter density.

The expected sensitivity is depicted by the dashed black line in Fig. 4 obtained by scaling the current limit by a factor of ten. A change in the absolute value of the magnetic field can be used to shift the carrier frequency and hence the sideband spectra to cover the blind spots of this analysis. The effective sensitivity of the future apparatus to axion-wind-like frequency modulations will be demonstrated by applying frequency modulations of the magnetic field.

## Acknowledgments

We acknowledge the excellent support provided by the PSI technical groups and by various services of the collaborating universities and research laboratories. In particular we acknowledge with gratitude the long term outstanding technical support by F. Burri and M. Meier. We thank the UCN source operation group BSQ for their support.

**Funding information**   We acknowledge financial support from the Swiss National Science Foundation through projects No. 117696, No. 137664, No. 144473, No. 157079, No. 163413, No. 172626, No. 126562, No. 169596 (all PSI), No. 181996 (Bern), and No. 172639, No. 200441 (both ETH). This work was also partly supported by the FWO Research Foundation Flanders. The work of YVS was supported by the Australian Research Council under the Discovery Early Career Researcher Award DE210101593.

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
