# Peer review of "Search for ultralight axion dark matter in a side-band analysis of a 199Hg free-spin precession signal"

_SciPost Physics, doi:SciPost Phys. 15, 058 (2023)_

## Round 1 · Referee Report · Hendrik Bekker (Referee 1) · 2023-3-9

Strengths

1) significantly improved sensitivity compared to previous work using a very similar method

2) Complementary to other searches

3) Well written

Weaknesses

1) More sensitive limits available in this mass range

2) Long coherence time of the sought-after signal does not seem to be exploited fully

Report

In the presented work, a $^{199}$Hg vapor-cell magnetometer is employed to search for axionlike dark matter particles in the mass range from $10^{−16}$ eV to $10^{−13}$ eV. The sought-after signal would appear as sidebands to the spin-precession signal of the mercury nuclei, similarly to the work done in reference [22]. Although the here obtained sensitivity is better when compared to that work, more recent work presents a significantly improved sensitivity (https://arxiv.org/abs/2209.03289).

The manuscript is well written with an especially clear introduction. Throughout the text, suitable references are made to other works and where relevant, the presented work is contrasted with existing ones. Unfortunately, the recent work by Lee et al, referred to above, is not present in this manuscript. This is an oversight which will have to be amended.

As pointed out in the manuscript, part of the relevance of this work is due to the stochastic behavior of the axionlike particles at the investigated low masses. For this, the reader could be referred to the following work for more details: https://www.nature.com/articles/s41467-021-27632-7

The applied experimental techniques are described clearly and appear to be scientifically sound. Figure 2 is a welcome addition that demonstrates the degree of experimental control. A mention of the average mercury vapor pressure could be useful in order to compare the sensitivity of this method to others. Because the sensitivity is directly proportional to the sample amount, it is also worth mentioning what the limit to the vapor pressure is.

The data analysis also seem to be sounds. However, as mentioned in one of the final paragraphs discussing possible improvement to the method, the sensitivity scaling in this work goes with $T^{1/4}$. By making use of the long coherence time of the dark matter signal at these low frequencies, this scaling can be significantly better when applying the coherent averaging technique where a $T^{1/2}$ scaling is possible as has been exploited in reference [22]. Why was that method not applied in this work?

Overall, the manuscript makes a good impression and the results are relevant to the community. After making the suggested minor changes and possibly improving the data analysis, this work should be suitable for publication.

Requested changes

1) Include the recent results in this mass range in Fig. 4 ((https://arxiv.org/abs/2209.03289)) and refer to https://www.nature.com/articles/s41467-021-27632-7

2) Provide details regarding the mercury vapor. Particularly, what was the vapor pressure?

3) Comment on why the coherent averaging technique was not employed.

4) optionally: Change the x-axis in Fig. 3 to be in log-scale, similarly to Fig. 4, so that details of the residuals at frequencies below 1 Hz can be studied by the reader.

  • validity: top
  • significance: ok
  • originality: ok
  • clarity: high
  • formatting: excellent
  • grammar: excellent

Author:  Philipp Schmidt-Wellenburg  on 2023-03-17  [id 3494]

(in reply to Report 1 by Hendrik Bekker on 2023-03-09)
Category:
answer to question

The authors would like to thank the referee for his work and his thoughtful comments. We acknowledge and partially share the referee's assessment of the strength and weaknesses of the manuscript.

Weaknesses: 1) More sensitive limits available in this mass range A: While a more sensitive limit in the published mass range for an ultra light axion exists, we emphasize the complementarity of our data and the method.

2) Long coherence time of the sought-after signal does not seem to be exploited fully

A: We would like to point out that the sensitivity of our measurements does in fact improve with the total measurement time T as \propto T^{1/2} rather than \propto T^{1/4}, since we operate in the temporally coherent regime (as far as the dark-matter field is concerned).
This is because T for us is shorter than the coherence time of the dark-matter field for the entire dark-matter mass range that we consider. We therefore fully exploit the time coherence of a potential signal in our data.

Requested changes: 1) Include the recent results in this mass range in Fig. 4 ((https://arxiv.org/abs/2209.03289)) and refer to https://www.nature.com/articles/s41467-021-27632-7 A: We agree and have added these references. We thank the referee for bringing the recent arXiv preprint to our attention. The results in this preprint replace the limits from old magnetometer data that we previously had in our Figure 4 as shown by the torquoise region. We have added a reference to this new preprint and updated the torquoise region in our Figure 4. We have added the Nature Communicantions paper to page 5 of our manuscript where we discuss the effects of stochastic fluctuations (where we previously referred to Ref. [38]).

2)  Provide details regarding the mercury vapor. Particularly, what was the vapor pressure?

A: We have provided the typical vapor pressue p_Hg = 4.4E-7mbar calculated using the mean opacity, the light absorption cross section of 2E-13cm^2 and the diameter of the precession chamber.

3)  Comment on why the coherent averaging technique was not employed.

A: The sensitivity of our measurements does in fact improve with the total measurement time T as \propto T^{1/2} rather than \propto T^{1/4} since we operate in the temporally coherent regime (as far as the dark-matter field is concerned). Our analysis exploits this fact by creating a single time synchronized data trace. This is because T for us is shorter than the coherence time of the dark-matter field for the entire dark-matter mass range that we consider. In future measurements, we intend to take data for a much longer time than the coherence time of the dark-matter field, in this case the sensitivity will only improve as \propto T^{1/4}.

4) optionally: Change the x-axis in Fig. 3 to be in log-scale, similarly to Fig. 4, so that details of the residuals at
frequencies below 1 Hz can be studied by the reader.

A: We prefer a presentation with a linear scale, as this better shows the residual frequencies above 1Hz.

---

## Round 1 · Referee Report · Anonymous (Referee 2) · 2023-3-28

Strengths

(1) Covers ultralight axions in a theoretically interesting mass range
(2) The approach while not quite as sensitive in its current form as the competition is nevertheless interesting and complementary to these other approaches - since nobody really knows exactly which approach will ultimately win (they all have their challenges) this method should be published

Weaknesses

(1) I was confused about their analysis of the scaling of the SNR with time - but I think after the other referee's report, this has been clarified

Report

I think this paper represents solid experimental work - while it doesn't exactly cover new experimental parameter space, given that many of these approaches are still being experimentally developed, I think it is worth publishing this paper. It will clearly stimulate other people to think along these directions and ultimately, it may well be that a combination of ideas serve to push the needle on these searches.

---

## Round 2 · Author Response

The authors would like to thank the referees for their work and their thoughtful comments.
We acknowledge and partially share the referee's assessment of the strength and weaknesses of the manuscript.
We have modified and changed several passages in the resubmitted version of the manuscript to address all comments and suggestions by the referees.

---

## Round 2 · List of Changes

1) Include the recent results in this mass range in Fig. 4 ((https://arxiv.org/abs/2209.03289))
and refer to https://www.nature.com/articles/s41467-021-27632-7
A: We agree and have added these references.
We thank the referee for bringing the recent arXiv preprint to our attention.
The results in this preprint replace the limits from old magnetometer data that we previously had in our Figure 4 as shown by the torquoise region.
We have added a reference to this new preprint and updated the torquoise region in our Figure 4.
We have added the Nature Communicantions paper to page 5 of our manuscript where we discuss the effects of stochastic fluctuations (where we previously referred to Ref. [38]).

2) Provide details regarding the mercury vapor. Particularly, what was the vapor pressure?
A: We have provided the typical partial pressure of mercury p_Hg = 4.4E-7mbar in the paragraph on the functioning of the mercuy magnetometer. We calculated the pr using the mean opacity, the light absorption cross section of 2E-13cm^2 and the diameter of the precession chamber.

3) Comment on why the coherent averaging technique was not employed.
A: The sensitivity of our measurements does in fact improve with the total measurement time T as \propto T^{1/2}
rather than \propto T^{1/4} since we operate in the temporally coherent regime (as far as the dark-matter field is concerned).
Our analysis exploits this fact by creating a single time synchronized data trace.
This is because T for us is shorter than the coherence time of the dark-matter field for the
entire dark-matter mass range that we consider.
In future measurements, we intend to take data for a much longer time than the coherence time of the dark-matter field,
in this case the sensitivity will only improve as \propto T^{1/4}.

4) optionally: Change the x-axis in Fig. 3 to be in log-scale, similarly to Fig. 4, so that details of the residuals at
frequencies below 1 Hz can be studied by the reader.
A: We prefer a presentation with a linear scale, as this better shows the residual frequencies above 1Hz.

You are currently on this page

Resubmission 2212.02403v2 on 3 April 2023

---

## Editorial Decision

published